# Inoculation with *Azospirillum brasilense* Strains AbV5 and AbV6 Increases Nutrition, Chlorophyll, and Leaf Yield of Hydroponic Lettuce

**DOI:** 10.3390/plants12173107

**Published:** 2023-08-29

**Authors:** Carlos Eduardo da Silva Oliveira, Arshad Jalal, Letícia Schenaide Vitória, Victoria Moraes Giolo, Thaissa Julyanne Soares Sena Oliveira, Jailson Vieira Aguilar, Liliane Santos de Camargos, Matheus Ribeiro Brambilla, Guilherme Carlos Fernandes, Pablo Forlan Vargas, Tiago Zoz, Marcelo Carvalho Minhoto Teixeira Filho

**Affiliations:** 1Department of Plant Protection, Rural Engineering and Soils, Faculty of Engineering, São Paulo State University—UNESP-FEIS, Ilha Solteira 15385-000, SP, Brazil; arshad.jalal@unesp.br (A.J.); leticia.schenaide@unesp.br (L.S.V.); victoria.giolo@unesp.br (V.M.G.); tjs.oliveira@unesp.br (T.J.S.S.O.); guilherme.carlos-fernandes@unesp.br (G.C.F.); 2Department of Biology and Zootechny, Faculty of Engineering, São Paulo State University—UNESP-FEIS, Ilha Solteira 15385-000, SP, Brazil; jailson.aguilar@unesp.br (J.V.A.); liliane.camargos@unesp.br (L.S.d.C.); matheus.brambilla@unesp.br (M.R.B.); 3Tropical Root and Starches Center (CERAT), Faculty of Agricultural Science, São Paulo State University—UNESP-FCA, Botucatu 18610-034, SP, Brazil; pablo.vargas@unesp.br; 4Unit of Mundo Novo, Department of Crop Science, State University of Mato Grosso do Sul—UEMS, Mundo Novo 79980-000, MS, Brazil; zoz@uems.br

**Keywords:** biofortification, carbon assimilation, *Lactuca sativa*, nutrients uptake, plant-growth-promoting bacteria

## Abstract

Inoculation with *Azospirillum brasilense* has promisingly increased plant yield and nutrient acquisition. The study aimed to estimate the dose of *A. brasilense* that increases yield, gas exchange, nutrition, and foliar nitrate reduction. The research was carried out in a greenhouse at Ilha Solteira, in a hydroponic system in randomized blocks with four replicates. The treatments consisted of doses of inoculation with *A. brasilense* strains AbV5 and AbV6 via nutrient solution (0, 8, 16, 32, and 64 mL 100 L^−1^). Inoculation with *A. brasilense* at calculated doses between 20 and 44 mL provided the highest fresh and dry mass of shoots and roots, number of leaves, and leaf yield. In addition, the calculated doses of inoculation with *A. brasilense* increased the accumulation of N, P, K, Ca, Mg, S, B, Fe, Mn, and Zn in shoots and roots, except the accumulation of Ca in roots. It also increased cell membrane integrity index (15%), relative water content (13%), net photosynthesis rate (85%), intracellular CO_2_ concentration (15%), total chlorophyll (46%), stomatal conductance (56%), transpiration (15%), and water use efficiency (59%). Hence, inoculation with *A. brasilense* at doses between 20 and 44 mL 100 L^−1^ is considered the best approach for increasing the growth, yield, accumulation of nutrients, and gas exchange of hydroponically grown iceberg lettuce.

## 1. Introduction

Lettuce is the most produced and consumed leafy vegetable in the world due to its desirable culinary characteristics for fresh consumption as salads [1]. Iceberg lettuce is a minimally processed vegetable that is increasingly in demand for consumption in restaurants and fast foods [2]. It has also been recognized as an important functional food due to its high contents of vitamins, minerals, and biologically active compounds, such as photosynthetic pigments and phenols [3]. Its cultivation in a hydroponic system has obtained more attention for healthy and quality production in a short duration [4], using a nutrient solution and protected environments to promote nutrient use efficiency for greater plant growth, nutrient accumulation, and even yield [5]. The higher concentration of nutrients (electrical conductivity) and time of exposure to the nutrient solution can increase water use efficiency, yield, and precocity of hydroponic lettuce [6]. Despite nutrient management, inoculation with plant-growth-promoting bacteria (PGPBs) is one of the sustainable alternatives that could increase nutrient use efficiency and accumulation by plants.

The use of PGPBs that improve growth, nutrient acquisition, water use, and yield gains has been extensively studied in different plant species [7]. In this regard, inoculation with *Azospirillum brasilense* has been associated with several mechanisms, such as increasing the production of phytohormones, siderophores, and secondary compounds as well as performing biological nitrogen fixation and promoting nutrient use efficiency, which promote plant growth [8]. It has also been reported that inoculation with *A. brasilense* strains AbV5 and AbV6 via foliar spray (at a dose of 1 mL L^−1^) increased the fresh mass of aerial parts and plant growth [9]. In addition, foliar inoculation with *A. brasilense* strains AbV5 and AbV6 provided greater leaf yield and accumulation of N, P, K, Ca, Zn, Cu, Mn, and Fe in the shoot of hydroponic lettuce, producing plants with higher quality [10]. Despite this, inoculation studies with other plant-growth-promoting bacteria such as *Arthrobacter pascens* strain BUAYN-122 and *Bacillus subtilis* strain BUABN-01 via nutrient solution promoted agronomic traits and nutrient acquisition of hydroponic lettuce and celery [4]. The mixed inoculation with *Azotobacter chroococcum*, *Azospirillum brasilense*, *Pseudomonas fluorescens*, and *Bacillus subtilis* in tomato seedlings increased fruit yield and nutrient uptake by plants in a hydroponic system [11].

Inoculation in a hydroponic system is a promising alternative option for higher nutrient uptake and productivity in lettuce and arugula cultivation. Hydroponic cultivation has reduced already as compared to soil cultivation while foliar inoculation has reduced pathogen infestation and pesticide applications, but it can increase the cost of production [12]. Thus, an inoculation option needs to be adapted to promote plant growth, health, and accumulation of nutrients in a more sustainable and economical manner.

Inoculation via nutrient solution has not been previously reported in a hydroponic system. A comparative effect of inoculation via foliar and nutrient solution with *A. brasilense* strains AbV5 and AbV6 in lettuce and arugula showed a similar effect in both cases [13]. Inoculation via nutrient solution with *A. brasilense* strains AbV5 and AbV6 is a new technique that can increase hydroponic production. It has been reported that *A. brasilense* strains AbV5 and AbV6 via nutrient solution increase the growth of roots and shoots and provide greater accumulation of nutrients in arugula and lettuce plants in a hydroponic system [13]. In addition, inoculation with *Pseudomonas fluorescens* strain CCTB03, *Bacillus subtilis* strain CCTB04, and *A. brasilense* strains AbV5 and AbV6 increased nutrient accumulation in shoots and roots, growth of shoots and roots, and biomass production of shoots and roots of arugula in a hydroponic system [14]. The previous studies verified the effectiveness of inoculation via nutrient solution; however, dose–response relationship and dose recommendations for *A. brasilense* strains AbV5 and AbV6 to promote greater productivity, nutritional enhancement, and photosynthetic efficiency still need to be defined.

The first reports on the inoculation of *A. brasilense* strains AbV5 and AbV6 via nutrient solution in a hydroponic system showed that there is an increase in root growth, shoot growth, accumulation of nutrients in shoots and roots, as well as photosynthetic efficiency. However, there still exists a research gap on the dose–response relationship of *A. brasilense* for greater plant growth, accumulation of nutrients, and plant physiology which may provide greater efficiency with this bacterium in lettuce plants. Therefore, the study aimed to verify the beneficial effects of different doses of *A. brasilense* strains AbV5 and AbV6 on the nutrition, physiology, growth, and leaf yield of lettuce cultivated in a hydroponic system.

## 2. Results

### 2.1. Productive Components

There was a significant effect of *A. brasilense* doses on the fresh and dry mass of shoot and roots, number of leaves, and fresh mass yield of iceberg lettuce leaves in a hydroponic system (Appendix A). The maximum number of leaves (20.5 leaves plant^−1^) was noted at the estimated dose of 40 mL 100 L^−1^ of inoculation with *A. brasilense* (Figure 1A). The yield of lettuce leaves was increased with inoculation of *A. brasilense* at a dose of 44 mL 100 L^−1^ of nutrient solution. The maximum yield was 7.88 kg m^−2^ of fresh iceberg lettuce (Figure 1B). The estimated optimal dose of 61 mL 100 L^−1^ of the inoculant provided the highest shoot fresh mass (424.5 g plant^−1^) of lettuce plants (Figure 1C). The root fresh and shoot dry matter of lettuce were linearly increased with increasing doses of *A. brasilense* via nutrient solution (Figure 1D,E). The calculated dose of 38 mL 100 L^−1^ of the inoculant provided greater root dry matter (2.1 g plant^−1^) of the lettuce plants (Figure 1F).

### 2.2. Inoculation with A. brasilense Improves Plant Physiology

There was a significant effect of *A. brasilense* doses on the net photosynthesis rate (A), internal CO_2_ concentration (Ci), stomatal conductance (gs), transpiration (E), water use efficiency (WUE), total chlorophyll content (ChlT), relative leaf water content (RWC), and leaf cell membrane integrity index (MII) (Appendix A). The highest A (16 µmol CO_2_ m^−2^ s^−1^) and WUE (1.47 mmol CO_2_ mol^−1^ H_2_O^−1^) were observed at the estimated dose of 32 mL 100 L^−1^ of inoculant (Figure 2A,E). The highest Ci (355 µmol CO_2_ mol^−1^ air^−1^) was observed at the maximum optimal dose of 20 mL 100 L^−1^ of the inoculant (Figure 2B). The estimated dose of 29 mL 100 L^−1^ of the inoculant was observed for the highest gs of 797 mol H_2_O m^−2^ s^−1^ (Figure 2C). The highest calculated E (10.9 mmol H_2_O m^−2^ s^−1^) was obtained at the estimated dose of 24 mL 100 L^−1^ of inoculant (Figure 2D). The highest concentration of total chlorophyll (1.42 mg g^−1^ FW) was noted at an estimated dose of 35 mL 100L^−1^ of *A. brasilense* (Figure 2F). The highest calculated RWC of the leaf was verified at a dose of 22 mL 100 L^−1^ of *A. brasilense*, with a maximum RWC of 93% in the leaves of hydroponic lettuce (Figure 2G). The MII of intact membranes of leaves was increased by 95% at a maximum calculated dose of 34 mL 100 L^−1^ of the inoculant, when compared with no-inoculation treatments (Figure 2H).

### 2.3. Macronutrient Accumulation in Lettuce Plants

The increasing doses of *A. brasilense* inoculation had a significant effect on the accumulation of N, P, K, Mg, S, B, Fe, Mn, and Zn in shoots and roots, and a significant effect on shoot Ca accumulation, while not significantly affecting the roots’ Ca accumulation of hydroponic lettuce (Appendix A). The highest calculated dose (64 mL 100 L^−1^) of *A. brasilense* impaired the nutritional balance of plants while inoculation with *A. brasilense* at the doses of 8, 16, and 32 mL 100 L^−1^ increased the accumulation of N in plants by 10, 12, and 6%, respectively, as compared to without inoculation treatments (Appendix A).

The highest shoot N accumulation (9.45 g m^−2^) of lettuce plants was observed at the maximum calculated dose of 37 mL 100 L^−1^ of *A. brasilense* via nutrient solution (Figure 3A). Higher shoot P accumulation (2.24 g m^−2^) was observed at a maximum calculated dose of 38 mL 100 L^−1^ of the inoculant (Figure 3B). The maximum accumulation of Ca and Mg in the shoot (4.48 and 1.18 g m^−2^) was observed at the estimated doses of 37 and 32 mL 100 L^−1^, respectively (Figure 3D,E). The maximum calculated shoot K accumulation (10.4 g m^−2^) was noted at the maximum optimal dose of 38 mL 100 L^−1^ of inoculant (Figure 3C). Shoot S accumulation of the lettuce was increased with increasing doses of *A. brasilense* inoculation in the nutrient solution in the hydroponic NFT system (Figure 3F).

The maximum root N accumulation (1.36 g m^−2^) in hydroponic lettuce plants was observed at the estimated dose of 34 mL 100 L^−1^ of the inoculant (Figure 3G). The calculated dose of 35 mL 100 L^−1^ provided the maximum root P accumulation (0.37 g m^−2^) (Figure 3H). The estimated dose of 41 mL 100 L^−1^ was noted with the maximum root K accumulation (1.02 g m^−2^) in the lettuce plants (Figure 3I). The maximum root S accumulation (0.41 g m^−2^) in the iceberg lettuce plants was noted at the estimated dose of 47 mL 100 L^−1^ of the inoculant (Figure 3L). There was no significant effect of *A. brasilense* doses on root Ca accumulation (Figure 3J). The greater root Mg accumulation (0.32 g m^−2^) was observed at the calculated dose of 35 mL 100 L^−1^ of the *A. brasilense* via nutrient solution while further increase in inoculation doses leads to the reduction (Figure 3K).

### 2.4. Micronutrient Accumulation in Lettuce Plants

The highest shoot B accumulation (6.48 mg m^−2^) was observed at the calculated dose of 37 mL 100 L^−1^ of the inoculant (Figure 4A). A linear increase in root B accumulation was verified with the increasing doses of *A. brasilense* inoculation via nutrient solution (Figure 4E). The maximum shoot Fe accumulation (423.27 mg m^−2^) was quantified at the estimated dose of 32 mL 100 L^−1^ of the inoculant (Figure 4D). Shoot Mn accumulation was linearly increased with increasing doses of *A. brasilense* via nutrient solution (Figure 4B). The highest accumulations of Fe (133.26 mg m^−2^) and Mn (10.42 mg m^−2^) in the roots were observed at the estimated dose of 37 mL 100 L^−1^ of the inoculant (Figure 4F,H). The maximum shoot Zn accumulation (50.24 mg m^−2^) was noted at a dose of 37 mL 100 L^−1^ of the inoculant (Figure 4C). In addition, the highest Zn accumulation (10.72 mg m^−2^) in the roots of hydroponic iceberg lettuce was verified at a dose of 35 mL 100 L^−1^ of the inoculant (Figure 4G).

## 3. Discussion

Inoculation with *A. brasilense* strains AbV5 and AbV6 has been reported as effective in increasing root growth in several plant species in soil cultivation systems. There are few reports of successful inoculation via nutrient solution with these strains due to their facultative endophytic nature [15]. In this context, the present study may explain the colonization ability of these bacterial strains in the hydroponic plant root system through greater fresh and dry matter of lettuce at a dose of 38 mL 100 L^−1^, compared to the non-inoculated plants (Figure 1B,D). The greater root growth might be due to the role of *A. brasilense* in the stimulation of phytohormone production, such as indole-3-acetic acid (IAA), which acts as a stimulator of root growth and consequently improves exploitation and absorption of water and nutrients [8]. A previous study also reported that inoculation with *A. brasilense* via nutrient solution at a dose of 10 mL 100 L^−1^ increased the fresh and dry mass of the roots of arugula grown in a hydroponic system [14]. The increase of IAA in the root zone resulting from the microorganism–plant interaction provides a greater number of secondary roots that are responsible for greater absorption of water and nutrients by the plants [16].

The fresh matter of lettuce is one of the most important traits of freshly marketable vegetables [6]. The current results showed that the shoot fresh and dry matter of hydroponic lettuce improved with increasing doses of *A. brasilense* when compared with non-inoculated plants (Figure 1A,C). It may be possible due to higher photosynthetic efficiency, assimilation of carbon, and nutrient acquisition by roots with the inoculation of *A. brasilense* [13]. It has been reported that inoculation with *A. brasilense* via nutrient solution at a dose of 10 mL 100 L^−1^ has increased the number of leaves, fresh and dry mass of shoots, and shoot growth of lettuce and arugula grown in a hydroponic system [13,14]. The yield and number of leaves of lettuce were increased with the doses of 44 and 20 mL 100 L^−1^ of the inoculant, respectively, in comparison to non-inoculated plants (Figure 1E,F). These results may be explained by the better plant nutrition and higher Ci, gs, and A in the current study (Figure 2A–C), which contribute to greater growth and biomass accumulation of the plants. Our results were supported by the previous findings that higher net assimilation of CO_2_ in the photosynthetic process and carbon accumulation allows the formation of new leaves and increased final yield [17].

Inoculation with *A. brasilense* increased RWC, MII, E, and WUE by 13, 15, 15, and 59% in the hydroponic lettuce leaves (Figure 2E–H). The integrity of leaf cell membranes also kept the metabolic and photosynthetic activities of plants in operation even in periods of high temperatures (Figure 2), as RWC kept the stomata open with gs 56% higher than in the absence of inoculation. It was previously reported that the water status of plant leaves can increase stomatal opening periods, which favors photosynthetic activity and net CO_2_ assimilation by plants, as well as improves water use efficiency and plant growth [18]. In addition, inoculation with an adequate dose of *A. brasilense* enhances the growth of lettuce plants by increasing nutrient acquisition for better uptake (Appendix A). Calcium is a nutrient that participates in the structural processes of plants, such as cell wall thickening, increases plant protection against pathogens, and enhances the restoration of cell membrane integrity [19]. In addition, the adequate supply and acquisition of B by plants are responsible for the formation and stabilization of primary cell walls, since cell wall integrity directly depends on B to carry out adequate signal transduction from the apoplast to the cytoplasm, thus critical for cell function [20]. Greater integrity of the cell wall membrane is essential for the growth and development of plant tissues to occur [21]. In addition, B and Ca promote the growth of new root hairs that effectively absorb water and nutrients for plant maintenance.

The accumulation of N, P, K, S, Ca, Mg, B, Cu, Zn, Fe, and Mn was improved by 39, 59, 51, 39, 57, 57, 60, 37, 48, 54, and 67%, respectively, with estimated optimal doses of inoculation with *A. brasilense* strains AbV5 and AbV6 as compared to non-inoculated plants (Figure 3 and Figure 4). The higher efficiency of micronutrient uptake and transport with the inoculation of *Azospirillum* sp. is mainly related to greater root growth, production of siderophores, secondary compounds, and phytohormones [22]. Inoculation via leaves and/or seeds with *A. brasilense* in soil-cultivated wheat has improved the absorption and translocation of Ca and Mg, consequently increasing the concentration of these nutrients in plant shoots [23]. They also provide greater uptake of Fe, Mn, B, and Zn in the root zone of plants and be translocated to the shoot, either by mass flow or by diffusion [24]. Inoculation with *A. brasilense* was previously reported to carry out food biofortification by increasing the absorption and concentration of Zn in grains of beans [25], Zn in corn and wheat [26,27], and Fe in corn [28]. It has been previously reported that inoculation with *A. brasilense* via leaves (dose 300 mL 250 L^−1^) in hydroponic cultivation favored root growth, which helped the plants achieve high accumulation of N, P, K, Ca, Zn, Cu, Mn, and Fe in lettuce [10]. Also, inoculation with *A. brasilense* via leaves (dose 300 mL 250 L^−1^) provided a higher concentration of N, P, K, Ca, Mg, and S in leaves of hydroponic arugula [9,29]. Inoculation with *A. brasilense* strains AbV5 and AbV6 (dose 10 mL 100 L^−1^) via nutrient solution promoted the growth of arugula plants in relation to non-inoculated treatments [14]. Therefore, our results described that growth and nutrient accumulation in hydroponic lettuce were improved with inoculation of *A. brasilense* strains AbV5 and AbV6 at doses between 39 and 50 mL (Figure 1, Figure 3 and Figure 4). The current results were supported by a previous study that verified the use of *A. brasilense* strains AbV5 and AbV6 increased growth and nutrient acquisition in hydroponic lettuce and arugula [13]. Plants adapt several mechanisms, such as biological nitrogen fixation, phosphate and zinc solubilization, greater root exploration, and induction of resistance to pathogens, when inoculated with *A. brasilense*. These could consequently increase the availability of N, P, Fe, Cu, Zn, and Mn in the soil, considering it a biofertilizer for greater nutrient uptake [30].

Our results showed that inoculation with *A. brasilense* strains AbV5 and AbV6 at an estimated dose of 35 mL 100 L^−1^ increased ChlT in the leaves of hydroponic lettuce (Figure 2F). Photosynthetic CO_2_ assimilation rates are favored by adequate K and Mg nutrition, increasing intracellular CO_2_ levels and leaf photosynthetic activity [31]. Due to the osmotic role of K in increasing leaf stomatal conductance (by regulating the guard cells), the lower K uptake impairs plant photosynthesis [32]. Plant growth and metabolism require the translocation of carbohydrates from photosynthetically active tissues. Sucrose loading is the main form of carbohydrate transport in plants, and deficiencies in K and Mg in the phloem can impair the efficient transport of carbon within the plant. An adequate supply of these two molecules enhances photoprotection [33]. In addition, in other studies, it was possible to verify an increase in the accumulation of K, Mg, and leaf chlorophyll content in lettuce and arugula inoculated with *A. brasilense* strains AbV5 and AbV6 in a hydroponic system [13]. The increase in photosynthetic activities due to the interaction of plants with PGPBs is related to the ability of these microorganisms to increase the production of extracellular polysaccharides, proteins, antibiotics, and glycoconjugates in plants, which are associated with greater photosynthetic activity in plants [34]. This strengthens our results regarding the effect of inoculation with *A. brasilense* on the interaction of leaf chlorophyll and the greater acquisition of nutrients.

The use of inoculation with *A. brasilense* has been highlighted as promoting plant growth and increasing crop yield and efficiency in nitrogen acquisition. The main mechanisms related to greater nitrogen acquisition caused by *A. brasilense* include greater root growth (which increases soil exploitation by roots), growth promotion (increased production of secondary compounds and phytohormones), and biological nitrogen fixation “BNF” (increased metabolism activity and yield by increasing carbon accumulation) [35]. The only source for uptake of available nitrogen in the hydroponic system is the roots, which indicated the highest accumulation of N (10, 12, and 6%) in the plants at the end of the cycle in relation to what was supplied via the nutrient solution. It can be explained by the process of biological nitrogen fixation provided by the inoculation with *A. brasilense* in the nutrient solution. In symbiosis with the plant, it sequesters N_2_ from the atmosphere, supplying it to the plants (it is suggestive to carry out BNF, and this is a hypothesis to be studied and proven, as BNF in lettuce hydroponic medium has not been reported before). The AbV5 and AbV6 strains of *A. brasilense* have, in their genetic sequencing, the expression of genes similar to *fixam* and *nif* that are responsible for the ability to perform atmospheric nitrogen fixation [36]. The use of these *A. brasilense* AbV5 and AbV6 strains may have greater efficiency in the use of nitrogen due to the ease of acquisition of ammonium molecules derived by the bacteria after the BNF process and their transport to plant tissues. Furthermore, inoculation with *A. brasilense* reduced the expression of the NF-YA gene, which is responsible for the high affinity of nitrate transporters (NO_3_^−^) from the roots to the shoot of the plants, inducing greater uptake of ammonium (NH_4_^+^) supplied to the plant by BNF [37]. The foremost acknowledged hypothesis with respect to the instrument of activity of *A. brasilense* is its development advancement, which incorporates nitrogen obsession and phytohormone, polyamine, and trehalose generation [38]. The mode of activity of *Azospirillum* is different, and the significance of each of these instruments can shift depending on climate, soil, and environmental conditions [39,40].

Based on the results of the current study, it is possible to highlight the importance of an inoculation dose that promotes productivity and improves the nutritional and physiological efficiency of hydroponic lettuce plants. In addition, it was possible to verify alterations in the nutrient solution with the inoculation of *A. brasilense* strains AbV5 and AbV6, which presupposes that this bacterium’s ability in interaction with the roots to fix atmospheric nitrogen may be associated. This hypothesis can be answered in future research using the ideal dose found in this study.

## 4. Materials and Methods

### 4.1. Environmental Characterization

The NFT hydroponic lettuce cultivation system was developed in protected cultivation with 30% shading at São Paulo State University (UNESP), Ilha Solteira-SP at 20°25′07″ S and 51°20′31″ W, and an altitude of 376 m. Meteorological data were collected from the inside of the greenhouse from an automatic meteorological station at UNESP between September 2nd and October 4th (Figure 5).

### 4.2. Experimental Delineation

The experiment was designed in randomized blocks with five replicates. Each experimental unit consisted of four lettuce plants, considering the four central channels of a hydroponic bench as a data collection area. The treatments consisted of doses of *Azospirillum brasilense* strains AbV5 and AbV6 (with a guaranteed count of 2 × 10^8^ CFU mL^−1^) at doses of 0 (non-inoculated control), 8, 16, 32, and 64 mL of the liquid inoculant for every 100 L of nutrient solution applied only on the day of transplanting the seedlings. The inoculant is a commercial product (Azotrop^®^) used in several countries and sold by Biotrop™ [41].

### 4.3. Experimental Characterization

The experiment was set up in an NFT system under individual benches. The characterization of the hydroponic system, periods of exposure of the plants to the nutrient solution, and flow rate of the nutrient solution used were described by Moreira et al. [10]. The nutrient concentrations used to formulate the nutrient solution (Hidrogood Fert Nacional, Hidrogood Fert Iron EDDHA and Calcium nitrate) were described by Moreira et al. [10].

The characteristics of the iceberg lettuce cultivar Angelina used were described by Moreira et al. [10] and Sakata [42]. The seedlings were developed in phenolic foam for 15 days and later transplanted to permanent benches of the NTF system, where they remained for 31 days until harvest.

The measurement and correction of conductivity and pH were performed daily in the morning; during these measurements, the electrical conductivity (EC) was readjusted to the EC determined for each cultivation bench, with the replacement of fertilizers if necessary. This process was performed on the 1st, 11th, and 21st days after transplanting (DAT). Weekly supplementation with 20 mg L^−1^ of K_2_O (KCl) was carried out for all benches to improve the water status of the plants because of the high temperatures in the region (determination carried out in previous studies since non-supplementation caused wilting in the plants in the period of highest daily heat). The initial electrical conductivity was 1.3 dS m^−1^, and these values were increased according to the stage of cultivation and its response to fertilization; so at 11th DAT, the EC was increased to 1.5 dS m^−1^, and at 21 DAT, it was raised to 1.7 dS m^−1^, which was maintained until harvest [14]. To maintain the pH between 6.0 and 6.5, sulfuric acid (25%) was used when the pH was above 6.5 and sodium hydroxide (25%) when the pH was below 6.0.

The replacement of electrical conductivity and pH was described according to the use of fertilizers (Figure 6A–E), and the effect of inoculation in the nutrient solution caused an increase in EC without the addition of nutrients (possibly due to biological nitrogen fixation “BNF”), thus avoiding nutrient replacement for a longer period. The daily elevations in electrical conductivity due to inoculation were described (Figure 6F).

### 4.4. Biometric and Yield Evaluations

The evaluations were carried out 31 days after transplanting the lettuce seedlings (harvest). Eight plants were collected and separated to quantify the number of leaves of each plant and the fresh mass of the root system and shoot (g), using a scale with an accuracy of 0.001 kg. Then, the material was sent for drying in an air-forced circulation oven at 60 °C for 72 h to obtain the dry mass of the root system and shoot in g, using an analytical scale with a precision of 0.001 g. Lettuce yield was based on the equation: fresh shoot weight in kg *X* plant population m^−2^ = yield in kg m^−2^. The plant population of 19.5 plants m^−2^ was used in the calculation.

### 4.5. Nutritional Evaluations

After drying, weighing, and grinding the plant materials in a Wiley-type mill, the concentrations of N, P, K, S, Ca, Mg, Fe, Mn, B, and Zn in the shoot and roots of lettuce were determined at harvest (31 days after transplanting) according to the methodology [43]. The accumulation of nutrients in the shoots and roots of the plants was calculated based on the respective values of dry mass and concentrations of nutrients. Using the equation: dry weight yield in kg m^−2^ X concentration of nutrients in g kg^−1^ or mg kg^−1^ = accumulation in g m^−2^ or mg m^−2^.

### 4.6. Physiological Evaluations

The evaluation of gas exchange was carried out on at least eight plants per treatment at the time of harvest, between 9 and 11 a.m., on fully expanded leaves located in the middle part of the plant, using an infrared gas analyzer (Infra-Red Gas Analyzer—IRGA, model CRS300). During this assessment, the internal CO_2_ concentration (Ci—μmol CO_2_ mol^−1^ air^−1^), stomatal conductance (gs—mmol of H_2_O m^−2^ s^−1^), transpiration (E—mmol of H_2_O m^−2^ s^−1^), net photosynthesis rate (A—μmol CO_2_ m^−2^ s^−1^), and water use efficiency (WUE—mmol CO_2_ mol^−1^ H_2_O^−1^) were measured.

The chlorophyll T (ChlT) content was determined using the extracting agent DMSO. Leaf tissue (50 mg) was cut into 1 mm fragments and incubated in 7 mL DMSO in the dark in a water bath at 65 °C for 30 min [44]. After readings in the spectrophotometer were made at 663 nm, the contents of the photosynthetic pigments were calculated and expressed in mg g^−1^ of fresh weight (FW) [45]. ChlT = (20.20 × ABS645) + (8.02 × ABS663).

In the final evaluation, the relative water content (RWC) and cell membrane integrity index (MII) were determined. The water status of plants was determined by the relative leaf water content (RWC). The RWC was calculated according to the equation: RWC (%) = [(FW − DW)/(TW − DW)] × 100. Ten leaf discs 25 mm in diameter were collected at 8:00 a.m. (at dawn) from four leaves of each plant and then immediately weighed to obtain the fresh weight (FW). To obtain the turgid weight (TW), leaf discs were submerged for 12 h in distilled water at 25 °C; after that, they were gently dried on a paper towel and weighed. The samples were then oven-dried at 75 °C for 24 h and weighed to determine the dry weight (DW). The leaf membrane stability index (MII) was measured by Lutts et al. [46] at harvest time. Using 10 leaf discs (25 mm in diameter) from 4 leaves of the plant, they were washed in distilled water, placed in tubes containing 50 mL of deionized water, and kept at 25 °C in a water bath for 6 h. The electrical conductivity of the solution was analyzed with an EC meter (EC1). Subsequently, the samples were placed in a water bath (100 °C) for 1 h, and the electrical conductivity (EC2) was evaluated after equilibration at 25 °C. The stability index of cell membranes was estimated according to the following equation: MII (%) = [1 − (EC1/EC2)] × 100.

### 4.7. Statistical Analysis

As the data of all variables presented a normal distribution and homogeneous variances (Shapiro–Wilk Test), they were subjected to analysis of variance. The F-test was used to assess the significance of the mean squares in the analysis of variance at the 5% probability level. The means relative to inoculant doses with *A. brasilense* were adjusted by regression analysis at the 5% probability using SigmaPlot 15 software [47].

## 5. Conclusions

The dose of 44 mL 100 L^−1^ of the inoculant of *A. brasilense* provided the highest yield of fresh leaves; that is, this is the dose indicated to achieve the greatest yield efficiency.

The use of doses of *A. brasilense* between 20 and 33 mL 100 L^−1^ improves the yield components and the accumulation of N, P, K, Ca, Mg, S, B, Fe, Mn, and Zn. Also, it was effective to perform biofortification with micronutrients in the leaves of iceberg lettuce plants in the hydroponic system.

Doses between 20 and 33 mL 100 L^−1^ of the inoculant with *A. brasilense* provided a better water status and gas exchange for lettuce plants, increasing photosynthetic, water use, and carbon assimilation efficiency.

Based on the results obtained, further research should be carried out to explore inoculation via nutrient solution with *A. brasilense* strains AbV5 and AbV6 interacting with other factors, such as reduction or increase in the electrical conductivity of the nutrient solution, reduction in the use of fertilizers, management of N in the nutrient solution, and increased precocity for harvesting and other plant species.

## Figures and Tables

**Figure 1 plants-12-03107-f001:**
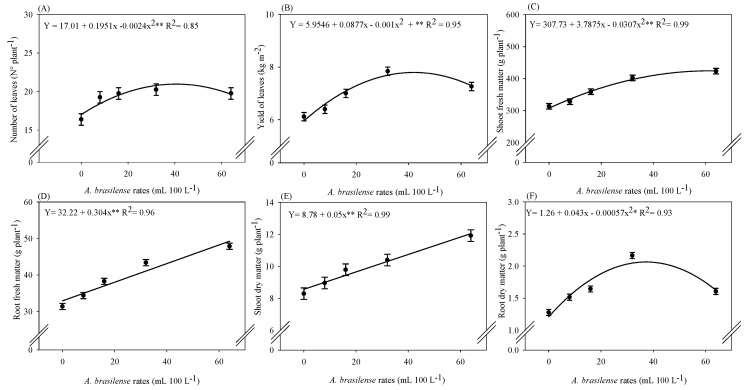
Number of leaves (**A**), yield of fresh leaves (**B**), fresh matter of shoot (**C**) and roots (**D**), and dry matter of shoot (**E**) and roots (**F**) of iceberg lettuce plants in NFT hydroponic system according to inoculation doses of *A. brasilense* via nutrient solution. The lines present in the significant regressions correspond to the linear adjustment or quadratic adjustment of the regression. The standard error is shown in ± error bars.

**Figure 2 plants-12-03107-f002:**
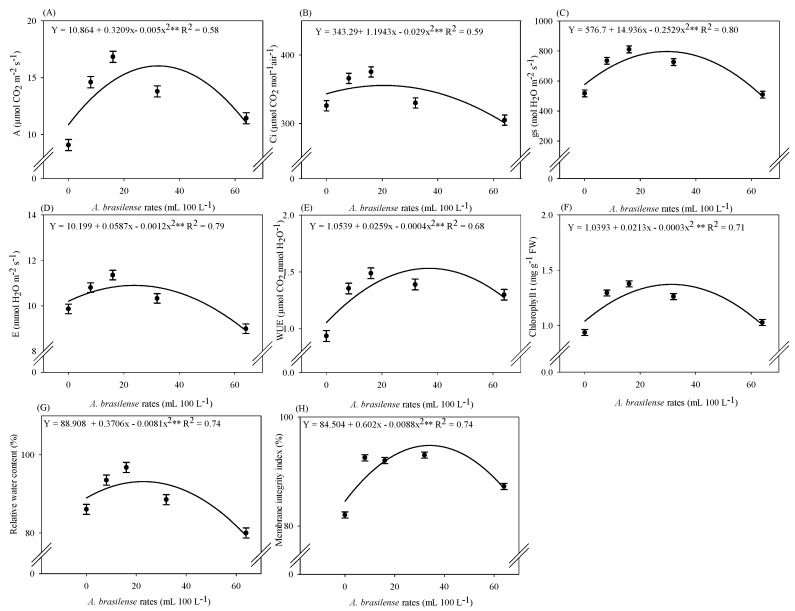
Net photosynthesis rate—E (**A**), internal CO_2_ concentration—Ci (**B**), stomatal conductance—gs (**C**), transpiration—E (**D**), water use efficiency—WUE (**E**), chlorophyll total content—ChlT (**F**), relative leaf water content (**G**), and leaf cell membrane integrity index (**H**) of iceberg lettuce plants in NFT hydroponic system according to the doses of *A. brasilense* via nutrient solution. The lines present in the significant regressions correspond to the linear adjustment or quadratic adjustment of the regression. The standard error is shown in ± error bars.

**Figure 3 plants-12-03107-f003:**
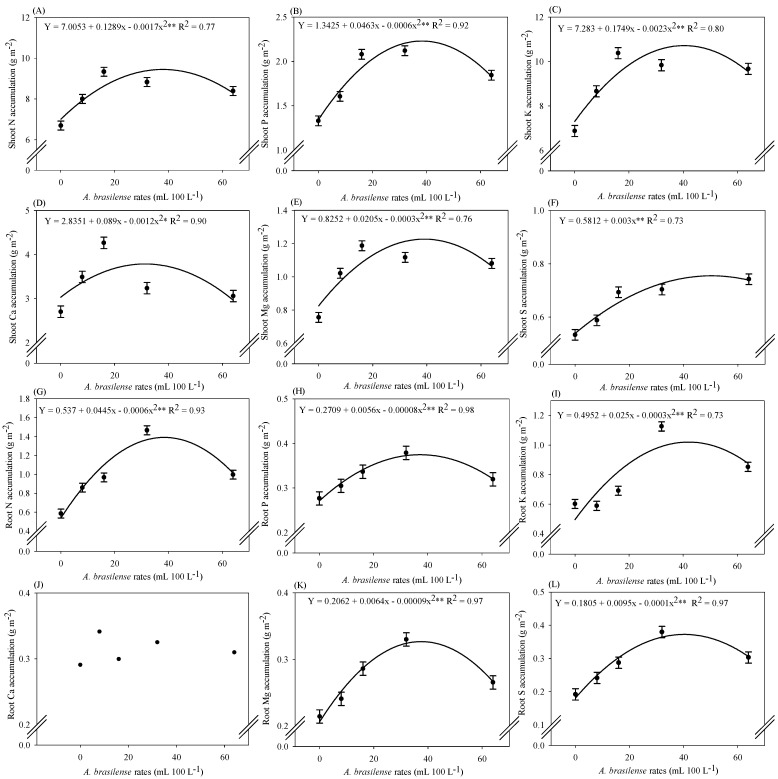
Accumulation of nitrogen in shoot (**A**) and roots (**G**), phosphorus in shoot (**B**) and roots (**H**), potassium in shoot (**C**) and roots (**I**), calcium in shoot (**D**) and roots (**J**), magnesium in shoot (**E**) and roots (**K**), and sulfur in shoot (**F**) and roots (**L**) of iceberg lettuce in NFT hydroponic system according to the doses of *A. brasilense* via nutrient solution. The lines present in the significant regressions correspond to the linear adjustment or quadratic adjustment of the regression. The standard error is shown in ± error bars.

**Figure 4 plants-12-03107-f004:**
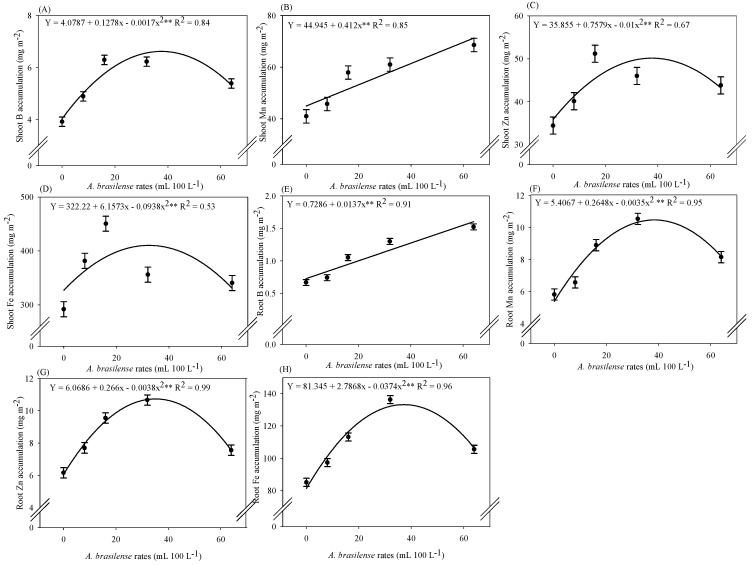
Accumulation of boron in the shoot (**A**) and roots (**E**), manganese in the shoot (**B**) and roots (**F**), zinc in the shoot (**C**) and roots (**G**), and iron in the shoot (**D**) and roots (**H**) of iceberg lettuce plants in the NFT hydroponic system according to the doses of *A. brasilense* via nutrient solution. The lines present in the significant regressions correspond to the linear adjustment or quadratic adjustment of the regression. The standard error is shown in ± error bars.

**Figure 5 plants-12-03107-f005:**
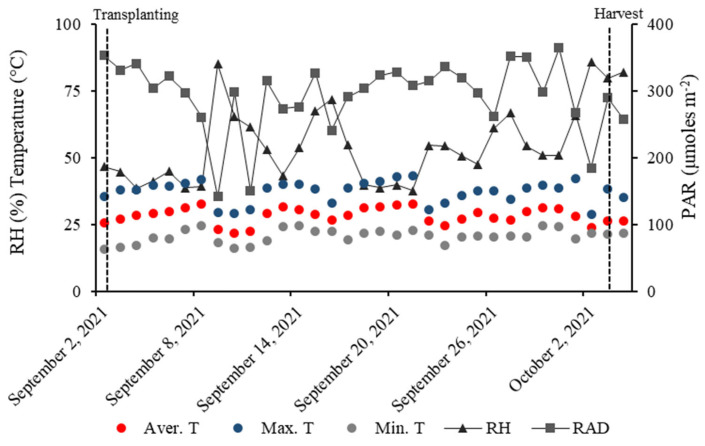
Relative air humidity (RH), maximum temperature (Max. T), average temperature (Aver. T), minimum temperature (Min T.), and PAR radiation (RAD) during the experiment.

**Figure 6 plants-12-03107-f006:**
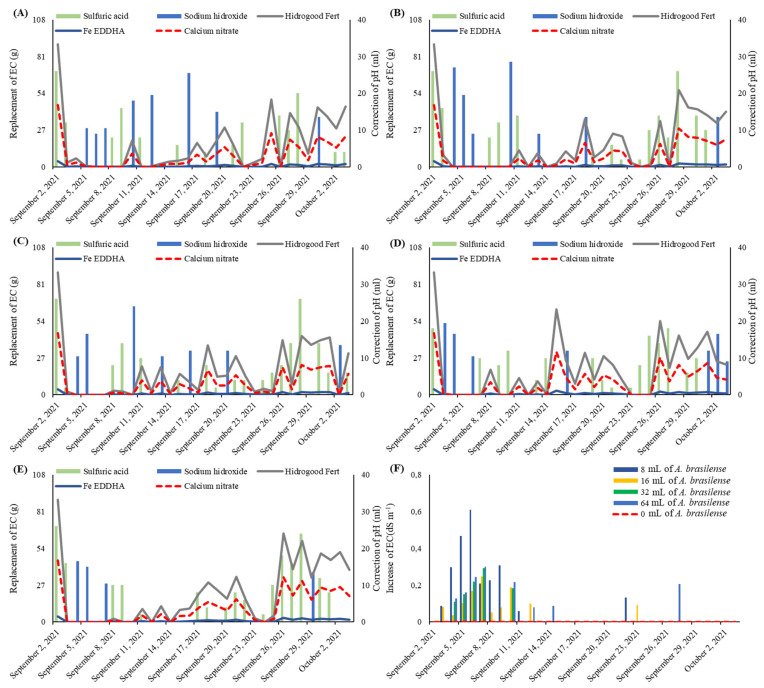
Replacement of pH and electrical conductivity (EC) in NFT hydroponic system in the control treatment (**A**), in the inoculation of *Azospirillum brasilense* via nutrient solution at a dose of 8 mL 100 L^−1^, (**B**), 16 mL 100 L^−1^ (**C**), 32 mL 100 L^−1^ (**D**), 64 mL 100 L^−1^ (**E**), and the increase in EC caused by inoculation according to each treatment (**F**) in the experimental period.

## Data Availability

The data will be available upon request from the corresponding and first authors.

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
