# Peer review of "Inoculation with Azospirillum brasilense Strains AbV5 and AbV6 Increases Nutrition, Chlorophyll, and Leaf Yield of Hydroponic Lettuce"

_plants, 2023, doi:10.3390/plants12173107_

Round 1

Reviewer 1 Report

This manuscript reports a study aimed at determining the ideal dose(s) of inoculation with Azospirillum brasilense to improve the fitness of hydroponic lettuce.

L26: “at a dose between 20 and 44 mL provided”. According to the previous sentence these concentration were not tested. Please explain.

L162: “This section may be divided by subheadings. It should provide a concise and precise description of the experimental results, their interpretation, as well as the experimental conclusions that can be drawn.” ???

L330: which sodium?

L390: “The means relative to inoculant doses with A. brasilense were adjusted by regression analysis at 5% probability”. This needs to be explained with more detail. Which software was used for the statistical analyses?

The font size in the figures is too small, making it difficult to interpret.

Figures 1 to 4: please explain in the legend the meaning of the lines.

Conclusions: what is/are the recommended dose(s) of application of A. brasilense?

The English language needs to be improved throughout the manuscript. I suggest that a native English speaker revises the manuscript.

Author Response

Response to Reviewer 1:

This manuscript reports a study aimed at determining the ideal dose(s) of inoculation with Azospirillum brasilense to improve the fitness of hydroponic lettuce.

L26: “at a dose between 20 and 44 mL provided”. According to the previous sentence these concentration were not tested. Please explain.

A: The doses used in the results and in the discussion of the study are based on the results obtained from the quadratic equation in the quadratic regression. The authors subjected data to the regression analysis, based on the linear and quadratic equations; our data showed quadratic curve therefore, the current doses between 20 and 40 ml were calculated on the basis of maximum increase point for each attribute.

L162: “This section may be divided by subheadings. It should provide a concise and precise description of the experimental results, their interpretation, as well as the experimental conclusions that can be drawn.” ???

A: We really apologize for this typographical mistake. When we were formatting our manuscript, we didn’t delete this section from the template, and it happened like that. We will take care next time and will try our best to not happen such a blunder gain. Thanks for highlighting.

L330: which sodium?

A: The authors thanks to the reviewer. The source of sodium was sodium hydroxide. It is also corrected in the main text.

L390: “The means relative to inoculant doses with A. brasilense were adjusted by regression analysis at 5% probability”. This needs to be explained with more detail. Which software was used for the statistical analyses?

A: Thanks, the SigmaPlot 15 statistical software was used. The information are inserted in the main text and the reference was cited.

The font size in the figures is too small, making it difficult to interpret.

A: The font size in the figures has improved. The resolution is also improved.

Figures 1 to 4: please explain in the legend the meaning of the lines.

A: The lines present in the significant regressions correspond to the linear adjustment or quadratic adjustment of the regression.

Conclusions: what is/are the recommended dose(s) of application of A. brasilense?

A: The first conclusion was the recommendation. “The dose of 44 mL 100 L-1 of the inoculant of A. brasilense provided the highest yield of fresh leaves; that this dose is indicated to achieve the greatest yield efficiency.”

Reviewer 2 Report

Manuscript plants-2496982 describes the results of cultivating lettuce plants inoculated with a suspension of strains Azospirillum brasilense Ab-V5 and Ab-V6 under hydroponic conditions. Despite all the material presented by the authors, the discussion of the results and the conclusions made by the authors do not seem convincing to me. For publication in the journal Plants, all parts of the manuscript must be significantly modified, including the illustrative material. In this regard, I suppose that the manuscript as submitted should be rejected.

Basic remarks:

1) The title and abstract of the manuscript should indicate the bacterial strains used for inoculation. On the one hand, not all strains of Azospirillum brasilense could promote the growth of lettuce, on the other hand, the taxonomic position of strains in modern microbiology is very flexible and strains can be reclassified. Therefore, I think that the indication of strain names is more important than their species name.

2) The Introduction section should be completely changed to mention the use of azospirilla and other bacteria in hydroponics and aeroponics in the cultivation of lettuce and other plants.

In addition, references 4, 5, 6 and 9 are given incorrectly. At the same time, half of the references in the introduction are articles by the authors. This can significantly reduce the interest in this publication of a wide range of readers.

Lines 60-62: the sentence “promising results were highlighted at the dose of 10 mL 100 L-1 of A. brasilense via nutrient solution as it increases the growth of roots and shoots in a hydroponic system” is unclear without source context and additional explanation.

3) The Results section should also be completely revised to include more experimental data. One-year experiment and eight plants for comparison of variants in terms of morphometric parameters are not enough. The graphs do not show measurement errors. The authors performed an analysis of variance, the values of least significant significance (LSD) should be indicated. I agree that in many variants the authors did observe a significant positive effect of bacterial inoculation (as stated in the title of the article). But the main text of the description of the results of the manuscript is reduced to comparing the effect of different doses of the inoculant. I do not see significant differences between the variants with different doses of bacteria on the figures.

In addition, the authors use a graphical method to find the optimal concentrations of bacteria for various plant parameters. I advise the authors not to use a bacterial concentration of 0 when plotting, as this variant is useful in determining the effect of the presence of bacteria in the system, but is not one of the inoculum dose variants. This variant (conc. = 0 ml 100 L-1) is biological control.

4) in the Discussion section, there are several false and dubious assumptions around which the discussion is built. For example, the first sentence of the section (lines 167-169) “however, there are no reports of successful inoculation via nutrient solution with this bacterium, as they are facultative endophytic bacteria and may not colonize the roots (Figure 1B and D).” – Firstly, there are many reports of successful inoculation through nutrient solution with azospiralla of various plants; secondly, not all strains of Azospirillum brasilense are shown to be endophytic, and even vice versa, only some strains are endophytes; thirdly, endophyte lifestyle is not an obstacle to the colonization of plant roots. Authors must provide references or justify the lack of references for such assumptions.

Further (lines 169-172) the authors report the positive effect they observed of inoculation with bacterial colonization of the roots, but no results of evaluation of colonization were given in the manuscript.

The authors' discussion of nitrogen uptake is highly questionable (lines 260-279). The assertion that strains of Azospirillum brasilense can fix nitrogen under hydroponic conditions, despite the presence of available forms of nitrogen in the nutrient solution, requires experimental confirmation.

5) sections Materials and methods also require corrections.

Section 4.2 - how was the bacterial suspension prepared? How was CFU determined? How accurate was the CFU value? Why was a two-strain culture used rather than pure cultures? What ratio of cells of the two strains was used for inoculation? How was it verified? Why were the doses of inoculants used in the work chosen? All doses used differ by a maximum of 8 times - this is less than one order of magnitude of concentration. I think that the error in preparing the bacterial suspension for inoculation will significantly interfere with the use of a certain dose.

Section 4.3. – How and with what accuracy were the pH measurements carried out? What is "sodium (25%)"? (line 330).

4.6 - line 371 - Incorrect formula for calculating the content of total chlorophyll.

Were bacterial counts carried out in the hydroponic system or on plant roots during and at the end of the experiment?

6) The Conclusion section should be completely changed to indicate the impact of the results obtained by the authors on any aspect of plant research.

Thus, I did not find convincing evidence in the manuscript that the observed positive effects are associated with the activity of the inoculated bacteria, and the differences in the effect of different doses of the inoculant will be reproduced when the experiment is independently repeated.

Author Response

Response to Reviewer 2:

Manuscript plants-2496982 describes the results of cultivating lettuce plants inoculated with a suspension of strains Azospirillum brasilense Ab-V5 and Ab-V6 under hydroponic conditions. Despite all the material presented by the authors, the discussion of the results and the conclusions made by the authors do not seem convincing to me. For publication in the journal Plants, all parts of the manuscript must be significantly modified, including the illustrative material. In this regard, I suppose that the manuscript as submitted should be rejected.

Basic remarks:

1) The title and abstract of the manuscript should indicate the bacterial strains used for inoculation. On the one hand, not all strains of Azospirillum brasilense could promote the growth of lettuce, on the other hand, the taxonomic position of strains in modern microbiology is very flexible and strains can be reclassified. Therefore, I think that the indication of strain names is more important than their species name.

A: The authors thanks to the reviewer of providing a critical review. We agree with the reviewer and include the information of A. brasilense strains in the title, abstract and throughout the text to clarify the identification of the strains easier for the reader. Hope this version met the expectations of the reviewer.  

2) The Introduction section should be completely changed to mention the use of azospirilla and other bacteria in hydroponics and aeroponics in the cultivation of lettuce and other plants.

In addition, references 4, 5, 6 and 9 are given incorrectly. At the same time, half of the references in the introduction are articles by the authors. This can significantly reduce the interest in this publication of a wide range of readers.

A: The introduction is revised with more detailed information about inoculant and strains. We are sorry for the self-citation; however, this is a kind of new study in the nutrient solution, and we worked and already published some other relative papers on the topic. That’s why we cited our papers but kindly check the references are related. We didn’t cite any non-relevant reference.

Lines 60-62: the sentence “promising results were highlighted at the dose of 10 mL 100 L-1 of A. brasilense via nutrient solution as it increases the growth of roots and shoots in a hydroponic system” is unclear without source context and additional explanation.

A: The requested changes were made in the introduction.

3) The Results section should also be completely revised to include more experimental data. One-year experiment and eight plants for comparison of variants in terms of morphometric parameters are not enough. The graphs do not show measurement errors. The authors performed an analysis of variance, the values of least significant significance (LSD) should be indicated. I agree that in many variants the authors did observe a significant positive effect of bacterial inoculation (as stated in the title of the article). But the main text of the description of the results of the manuscript is reduced to comparing the effect of different doses of the inoculant. I do not see significant differences between the variants with different doses of bacteria on the figures.

In addition, the authors use a graphical method to find the optimal concentrations of bacteria for various plant parameters. I advise the authors not to use a bacterial concentration of 0 when plotting, as this variant is useful in determining the effect of the presence of bacteria in the system, but is not one of the inoculum dose variants. This variant (conc. = 0 ml 100 L-1) is biological control.

A: Dear reviewer, the results are fully revised and organized. Full statistical data are available in supplementary material. In studies that intend to determine an efficient dose for a new cultivation system or new form of application, dose studies are performed, where dose 0 (used as control) is necessary to calculate the regression equations (These studies can be of doses of fertilizers, inoculants and pesticides). To use a polynomial regression model. It is necessary to have 4 doses and the dose 0 (control) will determine the quadratic equation for maximum efficient point or minimum point when it is a negative polynomial. Therefore, performing an average test does not apply to this kind of experiment model. When using regression, it is not possible to calculate the least significant difference (LSD), which is available when performing mean test like Tukey or Student.

Some published articles manage to show this applicability of the statistic.

 Amaral  Júnior,  W.  E.;  Esteves,  F.  R.;  Menezes  Filho,  A.  C.  P.;  Ventura,  M.  V.  A.  Evaluation  of Azospirillum  brasilensedose response on fresh and dry matter of shoot and root of corn plants. Revista de Agricultura Neotropical, Cassilândia-MS, v. 9, n. 4, e6993, oct./dec.2022. ISSN 2358-6303. DOI: https://doi.org/10.32404/rean.v9i4.6993

Bueno CB, Dos Santos RM, de Souza Buzo F, de Andrade da Silva MSR, Rigobelo EC. Effects of Chemical Fertilization and Microbial Inoculum on Bacillus subtilis Colonization in Soybean and Maize Plants. Front Microbiol. 2022 Jul 6;13:901157. https://doi.org/10.3389/fmicb.2022.901157

Cordeiro, C.F.d., Echer, F.R. Interactive Effects of Nitrogen-Fixing Bacteria Inoculation and Nitrogen Fertilization on Soybean Yield in Unfavorable Edaphoclimatic Environments. Sci Rep 9, 15606 (2019). https://doi.org/10.1038/s41598-019-52131-7

Galindo, F.S., Buzetti, S., Rodrigues, W.L. et al. Inoculation of Azospirillum brasilense associated with silicon as a liming source to improve nitrogen fertilization in wheat crops. Sci Rep 10, 6160 (2020). https://doi.org/10.1038/s41598-020-63095-4

Raimondi, M.A. et al. Dose-response curve to soil applied herbicides and susceptibility evaluation of different amaranthus species using model identity. Planta daninha, 33 (1), 2015. https://doi.org/10.1590/S0100-83582015000100016

4) in the Discussion section, there are several false and dubious assumptions around which the discussion is built. For example, the first sentence of the section (lines 167-169) “however, there are no reports of successful inoculation via nutrient solution with this bacterium, as they are facultative endophytic bacteria and may not colonize the roots (Figure 1B and D).” – Firstly, there are many reports of successful inoculation through nutrient solution with azospiralla of various plants; secondly, not all strains of Azospirillum brasilense are shown to be endophytic, and even vice versa, only some strains are endophytes; thirdly, endophyte lifestyle is not an obstacle to the colonization of plant roots. Authors must provide references or justify the lack of references for such assumptions.

Further (lines 169-172) the authors report the positive effect they observed of inoculation with bacterial colonization of the roots, but no results of evaluation of colonization were given in the manuscript.

A: Dear reviewer, these statements were based on studies carried out with the strains of A. brasilense used in this study. So I reformulated the sentence and cited the manuscript that evaluates the presence of these strains in corn roots. As reported, these two strains are facultative endophytic and it was a mistake to state that they all are, so this has been corrected.

The assumptions that have been proposed for the colonization of plant roots in the hydroponic system are based on the results of other studies that have verified this possibility.

The authors' discussion of nitrogen uptake is highly questionable (lines 260-279). The assertion that strains of Azospirillum brasilense can fix nitrogen under hydroponic conditions, despite the presence of available forms of nitrogen in the nutrient solution, requires experimental confirmation.

A: Dear reviewer, in several studies it was possible to highlight the ability of the AbV5 and AbV6 strains to fix nitrogen from the atmosphere (expression of genes like fixam and nif that are responsible for the atmospheric nitrogen fixation). In our studies, we were able to find an increase in the N levels of the nutrient solution throughout the lettuce cycle as described in the EC measurements and with the total accumulation of N in the lettuce plants subtracting, which was supplied and the rest in the nutrient solution. It was possible to find differences between these values, which was assumed to be due to BFN, as in other studies.

5) sections Materials and methods also require corrections.

Section 4.2 - how was the bacterial suspension prepared? How was CFU determined? How accurate was the CFU value? Why was a two-strain culture used rather than pure cultures? What ratio of cells of the two strains was used for inoculation? How was it verified? Why were the doses of inoculants used in the work chosen? All doses used differ by a maximum of 8 times - this is less than one order of magnitude of concentration. I think that the error in preparing the bacterial suspension for inoculation will significantly interfere with the use of a certain dose.

A: Dear Reviewer, the strains of bacteria used are commercial in South America, this research was carried out together with the company, and the bacteria were isolated, separated and sent to the UNESP Laboratory to carry out inoculation. The packaging came sealed with the seal of the Ministry of Agriculture of Brazil. The concentration of each strain is equal within the inoculant, totaling 2x108 mL-1 CFU. There are indications of these strains for cultivation with grasses and legumes, research with vegetables is being carried out together with several Brazilian, Chilean, Argentinean and Paraguayan research institutions. As it is a liquid inoculant widely used for seed inoculation, via in-furrow and foliar route, its solubility in water is effective and manages to provide safety in studies carried out in protected cultivation and in the field.

https://biotrop.com/produto/azotrop/

The combined use of the strains provides a balance between the AbV5 strain responsible for promoting greater plant growth by stimulating the production of phytohormones and has expression of genes similar to fixam and nif that are responsible for the ability to perform atmospheric nitrogen fixation, and the AbV6 strain responsible for the expression of genes similar responsible for conferring increased tolerance to abiotic stresses and to fixam and nif that are responsible for the ability to perform atmospheric nitrogen fixation.

Hungria, M.; Ribeiro, R.A.; Nogueira, M.A. Draft genome sequences of Azospirillum brasilense strains Ab-V5 and Ab-V6, commercially used in inoculants for grasses and legumes in Brazil. Genome Announcements, 2018, 6, e00393–18. https://doi.org/10.1128/genomea.00393-18

Fukami, J.; Nogueira, M.A.; Araujo, R.S.; Hungria, M. Accessing inoculation methods of maize and wheat with Azospirillum brasilense. AMB Express, 2016, 6, 1-13. https://doi.org/10.1186/s13568-015-0171-y

Section 4.3. – How and with what accuracy were the pH measurements carried out? What is "sodium (25%)"? (line 330).

A: A portable pH meter was used with an accuracy of 2 decimal places, 0.01, daily at 7 am as described in the text. Sodium hydroxide was used, corrected in the text.

4.6 - line 371 - Incorrect formula for calculating the content of total chlorophyll.

The equation is correct, the equation that makes the multiplication by the volume of acetone inserted is not considered, as it does not use acetone, using DMSO, as described by Wellburn, (1994).

Wellburn, A.R. The Spectral Determination of Chlorophylls a and b, as well as Total Carotenoids, Using Various Solvents with Spectrophotometers of Different Resolution. Journal of Plant Physiology, 1994, 144, 307-313. https://doi.org/10.1016/S0176-1617(11)81192-2

Being a method used in other research, and accepted worldwide.

BOMFIM, N. C. P. ; AGUILAR, J. V. ; PAIVA, W. S. ; SOUZA, Lucas Anjos de ; JUSTINO, G. C. ; Faria, G.A. et al. Phytostabilization by Leucaena leucocephala. South African Journal Of Botany, v. 138, p. 318-327, 2021. https://doi.org/10.1016/j.sajb.2021.01.013

AGUILAR, J. V.; LAPAZ, A. M.; SANCHES, C. V. ; YOSHIDA, C.H. P.; et al. Application of 2,4-D hormetic dose associated with the supply of nitrogen and nickel on cotton plants. Journal Of Environmental Science And Health Part B-Pesticides Food Contaminants And Agricultural Wastes, v. 56, p. 1, 2021. https://doi.org/10.1080/03601234.2021.1966280

BOMFIM, N. C. P. ; AGUILAR, J. V. ; FERREIRA, T. C. ; SOUZA, Lucas Anjos de ; de Camargos, L. S. . Could nitrogen compounds be indicators of tolerance to high doses of Cu and Fe in the cultivation of Leucaena leucocephala?. Plant Physiology And Biochemistry, p. 1, 2022. https://doi.org/10.1016/j.plaphy.2022.11.042

Were bacterial counts carried out in the hydroponic system or on plant roots during and at the end of the experiment?

A: Nutrient solution and roots were not collected to count for the number of bacteria present within the roots.

6) The Conclusion section should be completely changed to indicate the impact of the results obtained by the authors on any aspect of plant research.

Thus, I did not find convincing evidence in the manuscript that the observed positive effects are associated with the activity of the inoculated bacteria, and the differences in the effect of different doses of the inoculant will be reproduced when the experiment is independently repeated.

A: Dear reviewer, as it was carried out in greenhouse, with low climatic effect on the experiment, and the treatments were the only source of variation, then, I can say that there was an effect of the bacteria on the growth and accumulation of nutrients in the lettuce plants. This effect was not only reported in this study, as previously highlighted in your questions, in previous studies we were able to verify the effect of these strains via foliar and nutrient solution, and this was the starting point for developing new research to find the most efficient dose. to promote growth and nutrient accumulation.

Oliveira, C.E.S.; Gato, I.M.B.; Moreira, V.A.; Jalal, A.; Oliveira, T.J.S.S.; Oliveira, J.R.; Fernandes, G.C.; Teixeira Filho, M.C.M. Inoculation methods of Azospirillum brasilense in lettuce and arugula in the hydroponic system. Revista Brasileira de Engenharia Agrícola e Ambiental, 2023, 27, 653-662. http://dx.doi.org/10.1590/1807-1929/agriambi.v27n9p653-662

Moreira, V.D.A.; Oliveira, C.E.S.; Jalal, A.; Gato, I.M.B.; Oliveira, T.J.S.S.; Boleta, G.H.M.; Giolo, V.M.; Vitória, L.S.; Tamburi, K.V.; Teixeira Filho, M.C.M. Inoculation with Trichoderma harzianum and Azospirillum brasilense increases nutri-tion and yield of hydroponic lettuce. Archives of Microbiology, 2022, 204, e440. https://doi.org/10.1007/s00203-022-03047-w

Reviewer 3 Report

1- The title of this manuscript is interesting, but the total words for Abstract according to the MDPI format should not be more than 200 words.

2- At the first line of Abstract, A. brasilense should be written as full name (Azospirillum brasilense), and after that it can be written and used as A. brasielnse.

3- It is suggested that authors use the amount of increase of estimated parameters such as CO2 concentration, total chlorophyll, stomatal conductance, transpiration and water use efficiency inside the paranthesis.

4- Keywords are OK, but the first word of Biofortification (B), should be written as a small letter like other keywords of the manuscript (According to the MDPI format)

5- In Introduction, paragraphing is not OK, currently authors have used and being with 4 paragraphs which should be decreased to maximum two paragraphs, and each paragraph should start with new contents and points.

6- Line 63, why do authors write (11;12), delete ; and write (11,12).

7- If it is possible in the part of Results, increase the font size of columns of figures, for example for Figure 1, it is very difficult to read parameters in columns.

8- Again in line 96 and 97, in 2.2 section of Result, two paragraphs should be joined and it is not necessary to separate two paragraphs in this part.

9- Line 99, Please check and correct CO2 m-2air-1.

10- Figure 2 should be corrected exactly like Figure 1.

11- Line 13 in Figure 2, authors should write the abbreviation form of Azospirillum brasilense as A. brasilense.... This should be done in all part of the manuscript. 

12- Figure 3 should be corrected exactly like figure 1 and 2.

13- Figure 4 should be corrected as what I have mentioned in previous comments.

14- In the part of Discussion, paragraphing is not correct, as I did mention for Introduction, it should be revised and corrected.

15- In the part of Discussion, authors have not used enough references to compare the results of this experiment with the results of other experiments. They have clarified and illustrated the results of this experiment clearly, but the part of comparing the results of this trial with other and former experiments is not enough at all.

16- In the last paragraph which is the final conclusion of this manuscript, the authors should also clarify what sorts of experiments should be done in future researches which are related to this experiment.

17- Conclusion is not clear at all, and it is too short and it should be revised and re-written completely, it can not be accepted in this current format.

18- The references in this manuscript are not written and designed on the basis of format of MDPI, they are all should be corrected.

19- Just compare Reference 7 and 8, in reference 7, the authors have written https://doi.org/... but in number 8, they have written has DOI: ..... (???????)

20- Scientific names and latin names should be all Italics in the manuscript. For example in reference 40, Oryza sativa L. (rice) should be Italics.

21- The number of references for this manuscript is not enough, authors should more and better references from former published articles in MDPI, etc. to make the article rich.

The article needs Minor English revision.

Author Response

Response to Reviewer 3:

1- The title of this manuscript is interesting, but the total words for Abstract according to the MDPI format should not be more than 200 words.

A: Abstract adjustments have been made.

2- At the first line of Abstract, A. brasilense should be written as full name (Azospirillum brasilense), and after that it can be written and used as A. brasielnse.

A: Change has been made.

3- It is suggested that authors use the amount of increase of estimated parameters such as CO2 concentration, total chlorophyll, stomatal conductance, transpiration and water use efficiency inside the paranthesis.

A: Change has been made.

4- Keywords are OK, but the first word of Biofortification (B), should be written as a small letter like other keywords of the manuscript (According to the MDPI format)

A: Change has been made.

5- In Introduction, paragraphing is not OK, currently authors have used and being with 4 paragraphs which should be decreased to maximum two paragraphs, and each paragraph should start with new contents and points.

A: Change has been made.

6- Line 63, why do authors write (11;12), delete ; and write (11,12).

A: Other changes were made to the introduction, and then these references were changed.

7- If it is possible in the part of Results, increase the font size of columns of figures, for example for Figure 1, it is very difficult to read parameters in columns.

A: Enlargement of fonts within figures has been performed as suggested.

8- Again in line 96 and 97, in 2.2 section of Result, two paragraphs should be joined and it is not necessary to separate two paragraphs in this part.

A: Change has been made.

9- Line 99, Please check and correct CO2 m-2air-1.

A: Dear reviewer, the units of measurement have all been checked and entered as suggested by the manufacturer.

(CIRAS-3 Portable Photosynthesis System) URL: www.ppsystems.com  manual (https://ppsystems.com/download/technical_manuals/80097-1-CIRAS3_Operation_V200.pdf)

A Assimilation (µmol CO2 m-2 s-1)

Ci Sub-Stomatal CO2 Concentration (µmol CO2 mol-1 air-1)

gs Stomatal Conductance (mmol H2O m-2 s-1)

E Transpiration (mmol H2O m-2 s-1)

WUE Photosynthetic Water Use Efficiency (mmol CO2 mol-1 H2O)

10- Figure 2 should be corrected exactly like Figure 1.

A: Enlargement of fonts within figures has been performed as suggested.

11- Line 13 in Figure 2, authors should write the abbreviation form of Azospirillum brasilense as A. brasilense.... This should be done in all part of the manuscript.

A: Change has been made.

12- Figure 3 should be corrected exactly like figure 1 and 2.

A: Enlargement of fonts within figures has been performed as suggested.

13- Figure 4 should be corrected as what I have mentioned in previous comments.

A: Enlargement of fonts within figures has been performed as suggested.

14- In the part of Discussion, paragraphing is not correct, as I did mention for Introduction, it should be revised and corrected.

A: Change has been made.

15- In the part of Discussion, authors have not used enough references to compare the results of this experiment with the results of other experiments. They have clarified and illustrated the results of this experiment clearly, but the part of comparing the results of this trial with other and former experiments is not enough at all.

A: Dear Reviewer, there is not much previous research that addresses inoculation via nutrient solution with A. brasilense in hydroponics, the study that was cited and compared in the discussion was recently published on inoculation methods carried out by our research group.

16- In the last paragraph which is the final conclusion of this manuscript, the authors should also clarify what sorts of experiments should be done in future researches which are related to this experiment.

A: It was included in the discussion.

17- Conclusion is not clear at all, and it is too short and it should be revised and re-written completely, it can not be accepted in this current format.

A: A paragraph on future perspectives has been inserted and all results and discussion have been reformulated to make the conclusions succinct.

18- The references in this manuscript are not written and designed on the basis of format of MDPI, they are all should be corrected.

A: Adjusted.

19- Just compare Reference 7 and 8, in reference 7, the authors have written https://doi.org/... but in number 8, they have written has DOI: ..... (???????)

A: Dear Reviewer, Thanks for the annotations and all DOIs have been changed.

20- Scientific names and latin names should be all Italics in the manuscript. For example in reference 40, Oryza sativa L. (rice) should be Italics.

A: Dear Reviewer, Thanks for the annotations and all scientific names have been changed.

21- The number of references for this manuscript is not enough, authors should more and better references from former published articles in MDPI, etc. to make the article rich.

A: Dear Reviewer, the number of references in the list are increased.

There are no rules that suggest the exact number of references a research manuscript should contain, citing many articles from the MDPI platform in a manuscript to be published within a journal on the same platform is frowned upon by the international community and by the WEB OF SCIENCE that can penalize Plants for requesting this increase in citations.

Round 2

Reviewer 1 Report

All corrections have been made and the manuscript is now improved.

Author Response

A: Dear reviewer, thank you for your comments.

Reviewer 2 Report

The authors of the manuscript plants-2496982 have greatly improved the Introduction and Conclusion sections, and also responded to some of my minor remarks. However, the major problems of the manuscript, which I noted in the previous review, remained unresolved.

The most serious of these is the lack of a clear description of the bacterial inoculum. What were the plants treated with? As I understood from the answers of the authors (this is not in the text of the manuscript), that a commercial biological product was used in the form of a bacterial suspension, and the authors did not evaluate the amount of CFU/ml either at the beginning of the experiment or during it. Without knowing the differences in the content of bacterial cells in products of different commercial batches, it is impossible to assess the reliability of differences in variants that differ slightly in the dose of this biological product. I admit that the error in the content of bacterial cells in different parts of the inoculum may be greater than the differences in doses provided by the authors.

Another significant problem of this manuscript is the lack of independent replays of the experiment and the use of a low number of plants for analysis. Without proper statistical analysis, the results obtained in this way have no scientific significance. Such results may be useful to a biofertilizer manufacturer for promotional purposes, but they cannot be published as a scientific article, much less in the Plants.

For this reason, even after reviewing the authors' responses, I maintain my opinion that the manuscript should be rejected.

 In addition, I do not agree with the authors that

i) by using onto figures the regression, it is not possible to show the least significant difference (LSD), which can calculate with ANOVA;

ii) the activity of nif genes under model conditions can suggest nitrogen fixation by azospirilla in a hydroponic system - this requires experimental confirmation (for example, the acetylene reduction method);

iii) according to the formula indicated by them can correctly measure the content of total chlorophyll. To do this, you need to measure at two different wavelengths of light. And besides, the formula given by the authors is mathematically unclear.

Author Response

Response to Reviewer 2:

The authors of the manuscript plants-2496982 have greatly improved the Introduction and Conclusion sections, and also responded to some of my minor remarks. However, the major problems of the manuscript, which I noted in the previous review, remained unresolved.

The most serious of these is the lack of a clear description of the bacterial inoculum. What were the plants treated with? As I understood from the answers of the authors (this is not in the text of the manuscript), that a commercial biological product was used in the form of a bacterial suspension, and the authors did not evaluate the amount of CFU/ml either at the beginning of the experiment or during it. Without knowing the differences in the content of bacterial cells in products of different commercial batches, it is impossible to assess the reliability of differences in variants that differ slightly in the dose of this biological product. I admit that the error in the content of bacterial cells in different parts of the inoculum may be greater than the differences in doses provided by the authors.

A: Dear Reviewer, we have added the description in the introduction and highlighted for your kind reference in the paper last version. We really appreciate your microbiological approach about our manuscript however, we clearly mentioned in our methodology that the inoculant used in the current study was commercial and all the information about the product has provided by the company that’s why we didn’t evaluate the CFU/ml. This inoculant has used worldwide without any doubts. We just bought this inoculant from a bioinoculant’s marketing company, which claim its reliability and selling in several countries around the world. Our results, just support their claim about the inoculant and its role in growth promotion.

The reviewer also pointed out that the results found are restricted to product launches, We do not agree with this statement, as this inoculant is used in several studies that are published in several prestigious and well-known journals worldwide, such as: Plants, Agronomy, Microorganisms, Agronomy Journal, Scientific Reports, Archives of microbiology, Frontiers in Environmental Science, European Journal of Agronomy, Applied Soil Ecology, Nutrient Cycling In Agroecosystems, Frontiers in Plant Science. Therefore, I think the reliability of the inoculant should not be doubted and make it a base of rejection of the manuscript.  

Attached are several results obtained and published with this inoculant, including in a hydroponic system:

Oliveira, T.J.S.S.; Oliveira, C.E.S.; Jalal, A.; Gato, I.M.B.; Rauf, K.; Moreira, V.A.; Lima, B. H.; Vitória, L.S.; Giolo, V.M.; Teixeira Filho, M.C.M. Inoculation reduces nitrate accumulation and increases growth and nutrient accumulation in hydroponic arugula. Sci. Hortic. 2023, 320, 112213. https://doi.org/10.1016/j.scienta.2023.112213

Oliveira, C.E.S.; Gato, I.M.B.; Moreira, V.A.; Jalal, A.; Oliveira, T.J.S.S.; Oliveira, J.R.; Fernandes, G.C.; Teixeira Filho, M.C.M. Inoculation methods of Azospirillum brasilense in lettuce and arugula in the hydroponic system. Revista Brasileira de Engenharia Agrícola e Ambiental, 2023, 27, 653-662. http://dx.doi.org/10.1590/1807-1929/agriambi.v27n9p653-662

Moreira, V.D.A.; Oliveira, C.E.S.; Jalal, A.; Gato, I.M.B.; Oliveira, T.J.S.S.; Boleta, G.H.M.; Giolo, V.M.; Vitória, L.S.; Tamburi, K.V.; Teixeira Filho, M.C.M. Inoculation with Trichoderma harzianum and Azospirillum brasilense increases nutrition and yield of hydroponic lettuce. Archives of Microbiology, 2022, 204, e440. https://doi.org/10.1007/s00203-022-03047-w

Gato, I.M.B.; Oliveira, C.E.S.; Oliveira, T.J.S.S.; Jalal, A.; Moreira, V.A.; Giolo, V.M.; Vitoria, L.S.; Lima, B.H.; Vargas, P.F.; Teixeira Filho, M.C.M. Nutrition and yield of hydroponic arugula under inoculation of beneficial microorganisms. Hortic. Environ. Biote. 2023, 63, 1-12. https://doi.org/10.1007/s13580-022-00476-w

Amaral  Júnior,  W.  E.;  Esteves,  F.  R.;  Menezes  Filho,  A.  C.  P.;  Ventura,  M.  V.  A.  Evaluation  of Azospirillum  brasilensedose response on fresh and dry matter of shoot and root of corn plants. Revista de Agricultura Neotropical, Cassilândia-MS, v. 9, n. 4, e6993, oct./dec.2022. ISSN 2358-6303. DOI: https://doi.org/10.32404/rean.v9i4.6993

Galindo, Fernando Shintate; Bellotte, João Leonardo Miranda ; Santini, José Mateus Kondo ; Buzetti, Salatiér ; Rosa, Poliana Aparecida Leonel ; Jalal, Arshad ; Teixeira Filho, Marcelo Carvalho Minhoto . Zinc use efficiency of maize-wheat cropping after inoculation with Azospirillum brasilense. Nutrient Cycling In Agroecosystems, v. 1, p. 1, 2021.

SCUDELETTI, D. et al. Inoculation with Azospirillum brasilense as a strategy to enhance sugarcane biomass production and bioenergy potential. EUROPEAN JOURNAL OF AGRONOMY, v. 144, p. 126749, 2023.

GALINDO, F. S.; PAGLIARI, P. H. ; BUZETTI, S. ; RODRIGUES, W. L. ; FERNANDES, G. C. ; BIAGINI, A. L. C. ; TAVANTI, R. F. R. ; TEIXEIRA FILHO, M. C. M. . Nutrient availability affected by silicate and Azospirillum brasilense application in corn-wheat rotation. Agronomy Journal, v. 113, p. 4334-4347, 2021.

RONDINA, A.B. et al. Changes in root morphological traits in soybean co-inoculated with Bradyrhizobium spp. and Azospirillum brasilense or treated with A. brasilense exudates. BIOLOGY AND FERTILITY OF SOILS, v. 56, p. 537-549, 2020.

GALINDO, F. S.; BUZETTI, S. ; RODRIGUES, W. L. ; BOLETA, E. H. M. ; SILVA, V. M. ; TAVANTI, R. F. R. ; FERNANDES, G. C. ; BIAGINI, A. L. C. ; ROSA, P. A. L. ; TEIXEIRA FILHO, M. C. M. . Inoculation of Azospirillum brasilense associated with silicon as a liming source to improve nitrogen fertilization in wheat crops. Scientific Reports, v. 10, p. 6160, 2020.

ROSA, P. A. L.;  et al. Inoculation with Plant Growth-Promoting Bacteria to Reduce Phosphate Fertilization Requirement and Enhance Technological Quality and Yield of Sugarcane. Microorganisms, v. 10, p. 192, 2022.

Galindo, F.S., Buzetti, S., Rodrigues, W.L. et al. Inoculation of Azospirillum brasilense associated with silicon as a liming source to improve nitrogen fertilization in wheat crops. Sci Rep 10, 6160 (2020). https://doi.org/10.1038/s41598-020-63095-4

MORTINHO, E. S. et al.Co-Inoculations with Plant Growth-Promoting Bacteria in the Common Bean to Increase Efficiency of NPK Fertilization. Agronomy-Basel, v. 12, p. 1325, 2022.

JALAL, A. et al. Nanozinc and plant growth-promoting bacteria improve biochemical and metabolic attributes of maize in tropical Cerrado. Frontiers in Plant Science, v. 13, p. 1, 2023.

GALINDO, F. S.; PAGLIARI, P. H. ; FERNANDES, G. C. ; RODRIGUES, W. L. ; BOLETA, E. H. M. ; JALAL, A. ; CEU, E. G. O. ; LIMA, B. H. ; LAVRES JUNIOR, J. ; TEIXEIRA FILHO, M. C. M. . Improving sustainable field-grown wheat production with Azospirillum brasilense under tropical conditions: A potential tool for improving nitrogen management. FRONTIERS IN ENVIRONMENTAL SCIENCE, v. 10, p. 821628, 2022.

SILVA, E. R. et al. Can co-inoculation of Bradyrhizobium and Azospirillum alleviate adverse effects of drought stress on soybean (Glycine max L. Merrill.)?. ARCHIVES OF MICROBIOLOGY, p. 1-11, 2019.

Galindo, F. S.; Rodrigues, W. L. ; Fernandes, G. C. ; Boleta, E. H. M. ; Jalal, A. ; Rosa, P. A. L. ; Buzetti , S . ; Lavres Junior, J. ; Teixeira Filho, M. C. M. . Enhancing agronomic efficiency and maize grain yield with Azospirillum brasilense inoculation under Brazilian savannah conditions. EUROPEAN JOURNAL OF AGRONOMY, v. 134, p. 126471, 2022.

Galindo, F. S.; Silva, E. C. ; Pagliari, P. H. ; Fernandes, G. C. ; Rodrigues, W. L. ; Biagini, A. L. C. ; Baratella, E. B. ; Silva Junior, C. A. ; Moretti Neto, M. J. ; Silva, V. M. ; Muraoka, T. ; Teixeira Filho, Marcelo Carvalho Minhoto . Nitrogen recovery from fertilizer and use efficiency response to Bradyrhizobium sp. and Azospirillum brasilense combined with N rates in cowpea-wheat crop sequence. APPLIED SOIL ECOLOGY, v. 157, p. 103764, 2021.

Another significant problem of this manuscript is the lack of independent replays of the experiment and the use of a low number of plants for analysis. Without proper statistical analysis, the results obtained in this way have no scientific significance. Such results may be useful to a biofertilizer manufacturer for promotional purposes, but they cannot be published as a scientific article, much less in the Plants.

For this reason, even after reviewing the authors' responses, I maintain my opinion that the manuscript should be rejected.

 In addition, I do not agree with the authors that

  1. i) by using onto figures the regression, it is not possible to show the least significant difference (LSD), which can calculate with ANOVA;

A: The sample size used in this experiment was taken from the literature, we are not the one who just used 8 plants for data collection. It’s a standard way to collect data in the hydroponics. 

In addition, we don’t agree with the reviewer, how did he can say that we didn’t follow the proper statistics. We did follow the proper statistics but for the determining doses (quantitative), any statistician will go with regression not with LSD. We previously used ANOVA for means comparison however, regression analysis should be used to find a dose of product use that provides greater yield and efficiency.

Attached are references that highlight the use of regression to define the dose of a product in agriculture:

Quinn, G., Keough, M. Experimental Design and Data Analysis for Biologists. Cambridge, New York, 2002.

Gupta, et al. Statistical Analysis of Agricultural Experiments. ICAR-Indian Agricultural Statistics Research Institute. 2016.

Carvalho et al. A brief review of the classic methods of experimental statistics. Acta Schientiarum. Agronomy, 45,2023. https://doi.org/10.4025/actasciagron.v45i1.56882

Shimizu, G.D.; Gonçalves, L.S.A. AgroReg: main regression models in agricultural sciences implemented as an R Package. Sci. Agric. v.80, e20220041, 2023. http://doi.org/10.1590/1678-992X-2022-0041

Benjamin Doglas, Richard Kimwaga, Aloyce Mayo; A multiple regression model for prediction of optimal dose of Moringa Oleifera in faecal sludge dewatering. Water Practice and Technology 1 January 2022; 17 (1): 405–418. doi: https://doi.org/10.2166/wpt.2021.099

  1. ii) the activity of nif genes under model conditions can suggest nitrogen fixation by azospirilla in a hydroponic system - this requires experimental confirmation (for example, the acetylene reduction method);

A: Dear reviewer, it was previously reported that biological nitrogen fixation was assumed from the behavior of N accumulation and its supply by plants. However, at the end of the discussion of this item, the authors made it clear that further research should be carried out to prove this feat in the hydroponic system, something not previously reported.

iii) according to the formula indicated by them can correctly measure the content of total chlorophyll. To do this, you need to measure at two different wavelengths of light. And besides, the formula given by the authors is mathematically unclear.

A: Dear reviewer, the calculations for total chlorophyll do not need to use two bands of absorbance, this is necessary to perform the quantification of chlorophyll a and chlorophyll b, the model is attached:

The content of chlorophyll a (Chl a), b (Chl b) were determined using the extracting agent DMSO. Leaf tissue fresh weight (50 mg) was cut into 1-mm fragments and incubated in 7 mL DMSO in the dark in a water bath at 65 °C for 30 min (Hiscox and Israelstam, 1979). After readings in the spectrophotometer, the contents of the photosynthetic pigments were calculated and expressed in mg g-1 FW. According to the equations:

Chl a= (12.70×ABS663) ‒ (2.69×ABS645)

Chl b= (22.90×ABS645) ‒ (4.68×ABS663)

However, the total chlorophyll results support the physiological results obtained and provide greater reliability combined with data on leaf gas exchange.

Reviewer 3 Report

The article has revised very well, and the changes and the modifications have improved the article. However, there is just a suggestion in Paragraphing, authors have used many paragraphs in the text, for example at page 7 and 8, authors have used many paragraphs, and authors can decrease these paragraphs, and start each paragraph with new point.

In the Reference parts, some abbreviations are not correct, for example:  Reference 32, it should FEBS Lett (Not letters); Reference 28 should Hortic Environ Biotechnol (Not Biote);

Other parts of article is OK, I do suggest that authors also improve Discussion part with more references. After these minor revisions and suggestions, the article can be accepted for publication.

Author Response

Response to Reviewer 3:

The article has revised very well, and the changes and the modifications have improved the article. However, there is just a suggestion in Paragraphing, authors have used many paragraphs in the text, for example at page 7 and 8, authors have used many paragraphs, and authors can decrease these paragraphs, and start each paragraph with new point.

A: Dear reviewer, thank you for your comments. I changed the suggested item.

In the Reference parts, some abbreviations are not correct, for example:  Reference 32, it should FEBS Lett (Not letters); Reference 28 should Hortic Environ Biotechnol (Not Biote);

A: Dear reviewer, thank you for your comments. I followed the abbreviations I found from the web of science. I changed the suggested item.

https://images.webofknowledge.com/images/help/WOS/H_abrvjt.html

Other parts of article is OK, I do suggest that authors also improve Discussion part with more references. After these minor revisions and suggestions, the article can be accepted for publication.

A: Dear reviewer, thank you for your consideration, these suggestions have 

Round 3

Reviewer 2 Report

I do not agree with the authors that they sufficiently indicated that they used the commercial inoculant, while the phrases “There was a significant effect of A. brasilense doses on…” appear in the text. When these were not doses of bacteria, but doses of the commercial inoculant. Manufacturers of biofertilizers usually indicate how many live cells will be at the end of the shelf life, introducing significantly more bacteria into the suspension. In addition to bacteria, the preparations contain mineral salts and organic substances, which can also affect plants and influence the plant development or microbial community in a hydroponic system.

In addition, in two of the authors' replies, I did not find an answer to my question from review 1: "Why were these doses of inoculants used in the work chosen?". At the same time, it would be useful to indicate the rationale for the choice of doses in the manuscript.

Thus, I still find the description of the solution with which the plants were treated insufficient, but this is not a reason to reject publication. But the combination of this uncertainty with the lack of independent 1-2 repetitions of the experiment significantly reduce the scientific value of this publication. This is the reason for my decision to reject publication of this manuscript in Plants.

Some minor remarks:

1) Dear authors, adding bars (or LSD, or standard deviation, or confidence interval) to the figures for all the experimental points (mean values) showing the scatter of the data will be very useful to facilitate understanding of the results. The same regression analysis chart where the bars are 5% or 20% can lead to different conclusions. In addition, in the figures, the R2 value is between 0.53 and 0.99. This is not discussed in the text of the manuscript. How reliable are the results for determining the dose of the inoculant at R2 = 0.53?

2) Nitrogen fixation. If strains are so actively used in agrobiotechnology, they should have information under what conditions how much they fix atmospheric nitrogen. Otherwise, one can only hypothetically speak of their nitrogen fixation.

3) Total chlorophyll. A description of the methodology (Manuscript version 3) is given in lines 404-408. Only one reference [44] (Hiscox and Israelstam, 1979) is given at these lines. This is the reference to the extraction of pigments with DMSO. Hiscox and Israelstam used two wavelengths, 645 nm and 663 nm. Provide a reference (including in the manuscript) to the formula you used for determining total chlorophyll so that the readers can familiarize themselves with the theoretical justification for the formula. The fact is that in the formula there are two summands that have the same coefficient. So, I don't understand why the formula you gave is not simplified to ChlT = 28.22 × ABS663. Why do 3 math steps when you can get the same result in one step? In addition, a change in the Chla/Chlb ratio, which occurs with physiological changes in leaves, leads to changes in the optical density of solutions per unit of total chlorophyll. Therefore, I consider the above formula to be incorrect.

Author Response

I do not agree with the authors that they sufficiently indicated that they used the commercial inoculant, while the phrases “There was a significant effect of A. brasilense doses on…” appear in the text. When these were not doses of bacteria, but doses of the commercial inoculant. Manufacturers of biofertilizers usually indicate how many live cells will be at the end of the shelf life, introducing significantly more bacteria into the suspension. In addition to bacteria, the preparations contain mineral salts and organic substances, which can also affect plants and influence the plant development or microbial community in a hydroponic system.

A: Dear Reviewer, as previously indicated in another answer, the inoculant was provided by the company specifically for the research, so it was not purchased with the guarantee referred to at the end of validity, the inoculants supplied by the company in partnership with the University are sent with only the Viable cell count described in material and methods. I believe that it would be a problem, yes, to use the inoculant purchased directly from the company, as is the concern of the reviewer, however, at another point I explained that the university has an agreement with this company, where it supplies commercial and non-commercial inoculants with the bacteria count specifically for the experiment in question, we use it only during that week and later discard the inoculant, and when installing a new experiment, it is requested.

As it was described in the material and methods that an inoculant was used with the CFU, and the doses used were the inoculant, I do not see the need to change doses of A. brasilense for doses of inoculant containing A. brasilense, I believe that this is clear in the material and methods, and in other studies the strain and bacteria used are always reported, not the inoculant and how it was developed.

In addition, in two of the authors' replies, I did not find an answer to my question from review 1: "Why were these doses of inoculants used in the work chosen?". At the same time, it would be useful to indicate the rationale for the choice of doses in the manuscript.

A: There were no reports on the use of A. brasilense via nutrient solution in a hydroponic system, the doses used were based on the company's recommendations for application to the total area of ​​vegetables at a dose of 100 mL of the inoculant for 620 L of solution applied via irrigation or leaf. In this sense, the recommended dose would be 0.16 mL L-1 or 16 mL 100 L-1, so an application of 8, 16, 32 and 64 mL was carried out with the intention of obtaining the dose that promotes greater growth and yield of the lettuce in the hydroponic system.

Thus, I still find the description of the solution with which the plants were treated insufficient, but this is not a reason to reject publication. But the combination of this uncertainty with the lack of independent 1-2 repetitions of the experiment significantly reduce the scientific value of this publication. This is the reason for my decision to reject publication of this manuscript in Plants.

A: As in the items suggested for the reviewer, the standard error bar was inserted based on the statistical analysis in all regressions, to increase the reliability of the data presented, and doubts regarding the solution used as an inoculant are being resolved with each question. The research was carried out with all the criteria of rigor regarding the CFU of the inoculant and the strains of bacteria used, we carry out research with microorganisms in cooperation with private companies such as Biotrop and Koppert and other public institutions such as ESALQ-USP, UEMS, UFTPR, EMBRAPA , Antwerp University and Copenhagen University, studying the effect of several microorganisms in the most different conditions of soil, climate, planted species and stress, mainly with an interest in increasing crop yields with the use of bioinputs that can reduce environmental impacts.

Some minor remarks:

1) Dear authors, adding bars (or LSD, or standard deviation, or confidence interval) to the figures for all the experimental points (mean values) showing the scatter of the data will be very useful to facilitate understanding of the results. The same regression analysis chart where the bars are 5% or 20% can lead to different conclusions. In addition, in the figures, the R2 value is between 0.53 and 0.99. This is not discussed in the text of the manuscript. How reliable are the results for determining the dose of the inoculant at R2 = 0.53?

A: As in the items suggested for the reviewer, the standard error bar was inserted based on the statistical analysis in all regressions, the questioned adjustment of R2 = 0.53 of Fe accumulation in shoots, and the adjustment of R2 = 0.67 of Zn accumulation in shoots, are the only ones lower than R2 = 0.70, Both are micronutrients that are found in small amounts in the dry mass of plants, therefore, their quantification can normally be variable, however, the nutrient extraction methodology is widely used.

2) Nitrogen fixation. If strains are so actively used in agrobiotechnology, they should have information under what conditions how much they fix atmospheric nitrogen. Otherwise, one can only hypothetically speak of their nitrogen fixation.

A: Dear Revisor, all the studies that highlight the biological fixation of N by these strains of A. brasilense, were carried out in a culture system using soil, thus, as previously answered, our study generates a hypothesis that these strains can carry out biological N fixation in a soilless system (hydroponic) and confirming this feat are hypotheses for further studies. Our findings show that there was a greater accumulation of N in the plants in relation to the supply of N via nutrient solution, thus, there was the assumption that this extra N in the plants could come from the BNF. To prove this fact, there are ways to mark the 15N isotope in a hydroponic system, where all the N that is not marked in the plant comes from BNF.

3) Total chlorophyll. A description of the methodology (Manuscript version 3) is given in lines 404-408. Only one reference [44] (Hiscox and Israelstam, 1979) is given at these lines. This is the reference to the extraction of pigments with DMSO. Hiscox and Israelstam used two wavelengths, 645 nm and 663 nm. Provide a reference (including in the manuscript) to the formula you used for determining total chlorophyll so that the readers can familiarize themselves with the theoretical justification for the formula. The fact is that in the formula there are two summands that have the same coefficient. So, I don't understand why the formula you gave is not simplified to ChlT = 28.22 × ABS663. Why do 3 math steps when you can get the same result in one step? In addition, a change in the Chla/Chlb ratio, which occurs with physiological changes in leaves, leads to changes in the optical density of solutions per unit of total chlorophyll. Therefore, I consider the above formula to be incorrect.

A: Dear reviewer, Thank you for persisting that the total chlorophyll calculation was incorrect, previously the authors understood that the questions were related to the chlorophyll extraction method and its use. I had not understood the questions about total chlorophyll calculations, however, when checking we identified a minor error in the transcription of the formula, the correct one is ChlT = (20.20 x A645) + (8.02 x A663). It derives from the calculations proposed by Wellburn 1994 for photosynthetic pigments, where we calculated the fractions of chlorophyll A and chlorophyll B and not just total chlorophyll: Chla = (12.70 x A663) - (2.69x A645), Chlb = (22.90 x H645) - (4.68 x H663). However, in this study, we did not insert the data related to Chlorophyll a and b, due to the amount of data presented in the present study, the total chlorophyll was able to explain several physiological results allied to the data of leaf gaseous changes and the accumulation of nutrients.
